# Emotional and Behavioural Problems in Spanish University Students: Association with Lifestyle Habits and Mental Well-Being

**DOI:** 10.3390/healthcare12151482

**Published:** 2024-07-26

**Authors:** Gloria Tomás-Gallego, Raúl Jiménez Boraita, Javier Ortuño Sierra, Esther Gargallo Ibort, Josep María Dalmau Torres

**Affiliations:** 1University of La Rioja, 26006 Logroño, Spain; 2Department of Didactics of Physical Education and Health, Universidad Internacional de La Rioja, 26006 Logroño, Spain; raul.jimenez@unir.net; 3Department of Educational Sciences, University of La Rioja, 26006 Logroño, Spain; javier.ortuno@unirioja.es; 4Department of Didactics of Body Expression, University of La Rioja, 26006 Logroño, Spain; esther.gargallo@unirioja.es

**Keywords:** problem behaviours, emotional adaptation, mental health, habits, lifestyle, suicidal ideation, university

## Abstract

Emotional health represents a significant burden on the mental health of university students. Adapting to a new learning environment and facing increased academic pressure make this period particularly vulnerable for their emotional health and well-being. The objective of the study was to analyse the prevalence of emotional and behavioural problems in university students and their association with lifestyle habits, and mental and physical health indicators. A cross-sectional study was conducted on a sample of 1268 students (23.65 ± 7.84 years) from a university in northern Spain during November 2020 and March 2021. An online questionnaire was administered, comprising the self-report Strengths and Difficulties questionnaire, the Trait Meta-Mood Scale, the Rosenberg Self-Esteem Scale, the Satisfaction with Life Scale, the Perceived Stress Scale, the SENTIA-Brief Scale, the KIDMED questionnaire, the International Physical Activity Questionnaire-Short Form, the Alcohol Use Disorders Identification Test, and the Compulsive Internet Use Scale. 27.60% of students exhibited some form of emotional and behavioural problem. Students who did not present emotional and behavioural problems showed lower perceived stress, reduced suicidal behaviour and emotional intelligence deficits, as well as lower alcohol consumption and compulsive internet use (*p* < 0.001). Additionally, they reported higher engagement in physical activity and greater adherence to the Mediterranean diet (*p* < 0.001). The study shows that emotional and behavioural problems are recurring among university students, and given that modifiable psychosocial and lifestyle factors are associated with these issues, it underscores the need to develop multidisciplinary intervention strategies.

## 1. Introduction

An individual’s immediate surrounding environment is one of the main determinants of health. This makes it a priority when targeting improved health in the general population. Universities are ecosystems in which environmental, organizational and personal factors interact, with subsequent impact on the health and well-being of the university community. Furthermore, they represent organisations with great capacity for learning and knowledge generation, putting them in a prime position to develop interventions in concert with stakeholders and even incorporating immediate surrounding communities. The three groups comprising the university community, i.e., students, teachers–researchers and service workers, have specific health needs. The present article focuses on students [1,2,3,4].

The university student community brings together “emerging adults”, who range in age from 18 to 25 or 18 to 29 depending on the perspective of different authors and “adults”, i.e., all remaining university students. “Emerging adulthood” refers to a specific transitional period between adolescence and adulthood. It is characterised, specifically, by identity exploration, instability, self-focus, feelings of being in between and optimism/possibilities [5,6,7,8]. It is a critical moment during which previous childhood habits are consolidated or changed and those carried into adulthood, whether healthy or not, are consolidated [9,10,11]. The aforementioned characteristics, together with undertaking university studies, are significant factors in the emotional and mental health of this community, as the transition to university life involves significant life changes. For example, some students must leave their homes for the first time and distance themselves from their support networks, whilst also being faced with making decisions on their own. Thus, students comprise a vulnerable population at high risk of psychological distress and of developing or exacerbating mental and emotional health issues [12,13,14,15].

It is important to highlight that mental health includes emotional and behavioural aspects. Thus, emotional health is defined as a state of positive emotions and psychological well-being, constituting a complex framework that encompasses several fundamental components. These include negative emotions such as anger, fear and sadness; psychological well-being, which involves positive emotions, life satisfaction, and a sense of meaning; stress and self-efficacy, relating to the perception of stress and the ability to manage challenges; and social relationships, encompassing social support, companionship and social anxiety [16].

In a large international survey conducted at nineteen universities, covering eight countries and including fourteen thousand students, 35% of students met the diagnostic criteria for at least one common mental health issue [17,18]. Up to 50% of students worldwide are diagnosed with one or more mental disorders such as depression, anxiety and stress. Research such as that conducted by Ortuño-Sierra [19] shows that mental and emotional health problems are recurrent in the youth population, especially between the ages of 18 and 30. Worryingly, this situation has worsened following the COVID-19 pandemic [20,21,22,23], whilst university students are particularly vulnerable [24,25,26]. Even when students have similar health states to their non-university peers, they have been shown to exhibit more severe symptoms, failing to recover to the same pre-university level of health [15,27,28,29]. Students suffering from stress often believe that symptoms are a normal part of the university experience and so do not seek out help. Anxiety is one of the most common mental disorders amongst university students [30]. University students experience more suicidal thoughts and behaviours during their first onset than the same-age general population [31,32,33,34]. The present study applied the strengths and difficulties questionnaire (SDQ) to analyse behavioural and emotional problems in university students and examined their relationship with important risk factors for psychological disorders, namely, perceived stress, life satisfaction, self-esteem, emotional intelligence and suicidal behaviours.

Furthermore, these emotional and mental well-being are associated with recurrent somatic, psychological and behavioural disorders such as the consumption of toxic substances and internet addiction, poor sleep habits [35], sedentary behaviours, poor eating habits and dysfunctional relationships. This has a subsequent impact on academic performance, among other aspects [36,37,38]. Examples of this include the finding that higher Mediterranean diet adherence was correlated with lower depression risk, while higher perceived stress was related to lower fruit and vegetable intake. In contrast, good eating habits promote academic performance and quality of life, as well as mental and physical health status [39]. Students who engage in high amounts of physical activity, meet World Health Organization (WHO) physical activity guidelines or are athletes, whether at an amateur or professional level, report lower levels of depression and exhibit better health habits [40,41,42]. Finally, a lower grade point average has been found to be related to unhealthy mental outcomes and behavioural problems [23,43].

Spanish students follow the same trend. According to Ramón-Arbués [38] 34.5% of Spanish students suffer from stress and 42% from symptoms of anxiety and depression. And only one out of eight students with mental disorders receives mental health treatment [44]. Other studies carried out in Spanish universities have shown that students do not have healthy nutritional habits and do not meet the minimum physical activity requirements set by the WHO, both of which are predictors of mental health [37,45,46].

As discussed above, undergraduate students comprise part of a specific social group that often exhibits worse mental health states and unhealthy lifestyles. Due to the university cycle, students are susceptible to depression, stress, low self-esteem, academic and family pressure, early drop-out and, even, suicide, with risk being exacerbated at certain time periods such as during assessment periods [45,47]. Nevertheless, universities represent a privileged context from which to work on health promotion and perform actions that are meaningful, not only, to the immediate community, but, also, for society as a whole. Thus, the objective of the study was to analyse the prevalence of emotional and behavioural problems in university students and their association with lifestyle habits, and mental and physical health indicators. For this purpose, the following variables were analysed: emotional and behavioural problems, emotional intelligence, self-esteem, life satisfaction, perceived stress, suicidal behaviour, Mediterranean diet adherence, physical activity engagement, sedentary lifestyle, alcohol consumption and compulsive internet use.

## 2. Methods

### 2.1. Study Design and Participants

This university encompasses five faculties and two schools of higher education, with a total enrolment of 4408 students during the 2020–2021 academic year. Participants were recruited from various faculties and academic years using convenience sampling. Students enrolled in distance learning programs and those who did not understand Spanish (e.g., exchange students) were excluded, resulting in a remaining population of 4259 students. Initially, 2200 students were recruited for the study. After removing incomplete and questionnaires with random, pseudo-random, or dishonest responses, the final sample consisted of 1268 students (23.65 ± 7.84 years), ranging from 17 to 80 years old.

### 2.2. Procedure

The final instrument was created on the SurveyMonkey online platform and distributed to participants via institutional university email with the assistance and authorization of the Vice-Rectorate of Students. Emails contained a direct link which granted access to the web survey. Students responded individually from any device with an internet connection (i.e., mobile, tablet or computer). Responses could be provided from any location and only one response per user was possible. 

After obtaining approval from the Ethics Committee of the University, informed consent was obtained from all participants prior to their participation in the study. Confidentiality and anonymity of participant data were strictly maintained throughout the research process. Responses were gathered between November 2020 and March 2021.

### 2.3. Instruments

In the present study, a single instrument was elaborated by compiling a total of 11 validated tests and questionnaires alongside a number of socio-demographic questions. The different tests and questionnaires that made up the instrument are described below.

#### 2.3.1. Mental Health

The present study examined emotional and behavioural issues through application of a validated Spanish version of the self-report form for adults (+18 years), called the strengths and difficulties questionnaire (SDQ), conceived by Goodman [48]. This tool comprises 25 questions that a responded to using a three-point Likert scale. Questions are grouped into five subscales, namely, emotional problems, behavioural problems, peer problems, hyperactivity and prosocial behaviour, with each subscale comprising five items. At the same time as detecting disorders, the tool evaluates social strengths in children and adolescents. An overall score pertaining to disorders is generated by summing individual scores for all scales, with the exception of the prosocial scale. This produces possible overall scores that range between 0 and 40. Scores are categorised as normal (0–15), borderline (16–19) and abnormal (20–40). The strengths and difficulties questionnaire showed acceptable levels of reliability, α = 0.84 for the Total difficulties score, and ranging between 0.71 and 0.75 for the subscales [49].

The Trait Meta-Mood Scale (TMMS) is a self-report tool for measuring emotional intelligence that was designed by Salovey and colleagues [50]. It was developed to measure three cognitive components of emotional intelligence, namely, attention to feelings, clarity and repair. The original tool contains 48 items. In the present study, the abbreviated 24-item Spanish version elaborated by Fernandez-Berrocal and colleagues [51] was used. An equal number of items are framed positively and negatively, with responses given along a five-point Likert scale (i.e., from 1 = strongly disagree to 5 = strongly agree). Scores are calculated for each of the three components of emotional intelligence individually. Possible final scores range from 8 to 40 and outcomes are categorised according to gender. The reliability for the three aspects of perceived emotional intelligence in this scale is above 0.85 [51]. 

Self-esteem was assessed using a validated Spanish translation of the Rosenberg Self-Esteem Scale (RSES) [52,53], which measures respondents’ overall perceptions of their own self-esteem and worth. The tool is unidimensional and comprises 10 items, with five being positively framed and five being negatively framed. All items are rated using a four-point Likert-type scale (i.e., from 1 = totally disagree to 4 = totally agree). Final scores range between 10 and 40. The reliability index calculated for Spanish adults is 0.86 [54].

Global life satisfaction was measured in the present study using a validated Spanish translation of the Satisfaction with Life Scale (SWLS), originally conceived by Diener [55] and translated by Atienza and colleagues [56]. This instrument analyses global cognitive judgements of satisfaction with one’s own life through 5 items, such as “if I could live my life over, I would change almost nothing”. The scale comprises five items with responses given along a five-point Likert scale ranging from strongly disagree (1) to strongly agree (5). Potential final scores range from 5 to 35. Higher scores indicate greater life satisfaction. The reliability of this scale calculated using Cronbach’s Alpha coefficient was equal to 0.836 [57].

Perceived stress was measured using the Spanish adaptation, elaborated by Remor [58], of the Perceived Stress Scale (PSS), originally conceived by Cohen et al. [59]. This instrument asks about feelings and thoughts experienced during the month prior to instrument administration. It comprises 14 items which are rated along a five-point Likert scale (i.e., from 0 = never to 4 = very often) and, in each case, respondents are asked how often they felt a certain way. Total scores are obtained by, first, recoding scores given to items 4, 5, 6, 7, 9, 10 and 13 (in the following sense: 0 = 4, 1 = 3, 2 = 2, 3 = 1 and 4 = 0) and, then, summing scores for all 14 items, producing final scores in the range of 0 to 56. Higher scores indicate higher levels of perceived stress. The reliability of the Spanish short version was adequate, α = 0.82 [58].

The SENTIA-Brief Scale [60] assesses suicidal behaviour in accordance with five statements pertaining to individuals’ thoughts and feelings over the six months prior to questionnaire administration. Response options are dichotomous (yes or no) and total scores are obtained by attributing all affirmative responses with a value of zero and summing them together. Possible total scores range between 0 and 5. Higher scores indicate greater severity or risk of suicide. This scale must be used in combination with other assessment tools and alongside individual biopsychosocial analysis before making a clinical judgment. The reliability was equal to 0.97 Ω [60].

#### 2.3.2. Nutrition

The KIDMED questionnaire [61] measures Mediterranean diet adherence. This tool comprises sixteen items that assess whether or not patterns inherent to the Mediterranean diet are followed. Responses are dichotomous (yes or no). Affirmative responses to positively framed questions are coded as one, whilst affirmative responses to negatively framed questions are coded as minus one. Final scores are then summed, meaning that possible final score ranges between four and 12. Higher scores reflect greater Mediterranean diet adherence. This questionnaire has a reliability of Kappa value ranging from 0.504 to 0.849 [62].

#### 2.3.3. Physical Activity and Sedentary Behaviour

Physical activity (PA) engagement and sedentary habits were estimated through the Spanish version [63,64] of the self-report International Physical Activity Questionnaire-Short Form (IPAQ-SF). This measure assesses the intensity and type of PA engaged in, i.e., vigorous, moderate and walking, and sitting time, during the seven days prior to administration. For each type of activity, frequency and duration were collected. Total scores are calculated by summing the duration (in minutes) and frequency (days) of walking, and moderate-intensity and vigorous-intensity physical activity. Volume of physical activity was calculated according to METs (metabolic equivalent). In order to classify PA engagement, final scores are then categorised as high (total PA engagement of between 1500 and 3000 MET-minutes/week), moderate (between 600 and 1500 MET-minutes/week) and low (all remaining individuals who do not meet criteria for the other categories). In addition, this instrument assesses sedentary behaviour through a single dichotomous item. This item considers respondents to lead a sedentary lifestyle if they report engaging in more than 6 h per day of sedentary activities. The reliability data for the IPAQ short questionnaires are above 0.65 Spearman’s correlation coefficient [63].

#### 2.3.4. Toxic Consumptions

In order to identify whether students exhibited hazardous and harmful patterns of alcohol consumption, the Spanish-translated version for university students [65] of the standardized tool developed by the World Health Organization (WHO), the Alcohol Use Disorders Identification Test (AUDIT), was used. This tool comprises ten questions regarding the quantity, frequency and consequences of alcohol consumption. Scores for all items are summed, with possible overall scores ranging from 0 to 40, and a determination is made regarding whether alcohol consumption is a risk or reflective of dependent behaviour. Regarding the reliability of the Spanish version in university students, Spearman’s correlation coefficient was 0.87 [65].

In the present study, the Spanish version used in EDADES 2017 of the Compulsive Internet Use Scale (CIUS) [66] was employed to examine problematic internet use (UPI). This tool measures five dimensions: loss of control (items 1, 2, 5, and 9), preoccupation (items 4, 6, and 7), withdrawal symptoms (item 14), coping or mood modification (items 12 and 13) and, inter- and intrapersonal conflict (items 3, 8, 10, and 11). The CIUS consists of 14 items that are rated on a five-point Likert scale. Overall scores are obtained by summing scores for all individual items. Respondents are considered to exhibit compulsive internet use in cases in which CIUS scores are equal to or higher than 28. This scale has a degree of reliability equal to 0.91 of McDonald’s Omega [67].

Finally, the present study included two randomly interspersed pairs of questions from the Oviedo Infrequency Scale (INF-OV) within the compilation of all other survey measures. Fonseca-Pedrero and colleagues [68] conceived this tool as a means of detecting respondents who provide random, pseudorandom or dishonest responses to a questionnaire or test. It comprises 12 self-report items which are designed to have one obvious correct response from dichotomous response options (0 = yes; 1 = no). Students providing two or more incorrect responses on this test were removed from further analysis. Based on this, a total of 14 participants were excluded.

### 2.4. Statistical Analysis

Initially, on the one hand, quantitative variables were analysed according to means and standard deviations and, on the other hand, qualitative variables were analysed according to frequencies.

Subsequently, normality and homoscedasticity of the data were assessed for all variables, applying the Kolmogorov–Smirnov test with Lilliefors correction and the Levene test. Data for all variables were found to be non-normally distributed. Consequently, non-parametric tests were conducted for data analysis, i.e., Kruskal–Wallis test and Spearman correlations, with the aim of analysing the correlation between selected variables.

Finally, multiple linear regression was performed to identify factors associated with the emotional and behavioural difficulties identified by the strengths and difficulties questionnaire. Variables entered into the model were perceived stress, suicidal behaviour, self-esteem, life satisfaction, emotional intelligence, Mediterranean diet adherence, physical activity engagement, sedentary behaviour, alcohol consumption and compulsive internet use. Backward entry was used with variables with *p*-values < 0.05 being retained in the final model. 

Collected data were analysed with version 29 of IBM SPSS Statistics. Statistical significance was established at *p* < 0.05.

## 3. Results

Table 1 shows the sociodemographic data. The sample consisted of 64.9% women and 35.1% men. For this study, two groups were formed according to age, following Arnett’s social theory [5,6]: group 1, considered as “emerging adults”, with an age range between 17 and 25 years old (80.5%); and group 2, considered as “adults”, with n = 247 and an age range between 26 and 80 years old (19.5%). Additionally, 24.4% were employed, and their average monthly income was between EUR 0 and 499 in 60% of cases. 

Table 2 presents outcomes regarding perceived stress, suicidal behaviour, self-esteem, life satisfaction and emotional intelligence in university students as a function of emotional and behavioural symptomatology, classified according to standards defined by the strengths and difficulties questionnaire scoring outline [48]. In the present study, 72.40% of students exhibit capacities and difficulties that are considered to be normal and 16.40% of students exhibit borderline behaviour, whilst 11.20% are considered to belong to the abnormal group. Analysis revealed that perceived stress, suicidal behaviour and the attention to feelings component of emotional intelligence were significantly lower among students with normal emotional and behavioural symptomatology than among those with borderline or pathological symptomatology (*p* < 0.001). Likewise, indices describing self-esteem and life satisfaction, in addition to the clarity and repair components of emotional intelligence, were higher for students with normal emotional and behavioural symptomatology (*p* < 0.001).

Table 3 presents outcomes pertaining to physical activity engagement, sedentary behaviour, Mediterranean diet adherence, alcohol consumption and compulsive internet use according to the emotional and behavioural difficulties experienced by university students. Analysis revealed that students with normal emotional and behavioural symptomatology presented higher levels of physical activity engagement and Mediterranean diet adherence (*p* < 0.001), as well as lower sedentary behaviour engagement (*p* = 0.007), alcohol consumption and compulsive internet use compared to those with borderline or abnormal emotional and behavioural symptomatology.

Associations between emotional and behavioural symptomatology and all other examined variables are presented in Table 4. It can be seen that emotional and behavioural difficulties are positively correlated with perceived stress, suicidal behaviour, the attention to feelings component of emotional intelligence, sedentary behaviour, alcohol consumption and compulsive internet use. On the other hand, difficulties are negatively correlated with self-esteem, life satisfaction, physical activity engagement and Mediterranean diet adherence, in addition to the clarity and repair components of emotional intelligence.

Finally, Table 5 presents multiple linear regression outcomes pertaining to emotional and behavioural symptoms. Outcomes reveal that higher perceived stress, suicidal behaviour, attention to feelings, emotional intelligence and compulsive internet use, as well as lower self-esteem, satisfaction with life, emotional clarity and repair, and Mediterranean diet adherence were associated with increased emotional and behavioural difficulties. Further, the overall model explained 49.2% of the variance in the measured variables.

## 4. Discussion

Firstly, the present study reveals that 27.80% of students attending a university in northern Spain exhibit mental and behavioural disorders, of which 11.20% are severe. These percentages are close to what Goodman [48] expected when the original measure was conceived. The original study, conducted with a UK community sample, suggested that around 80% of the population could be expected to be rated as “normal” with regards to mental health. This finding is also broadly consistent with earlier work conducted with university students in that it documents a high prevalence of common mental disorders before, during and after the COVID-19 pandemic. Following administration of a series of surveys at 19 universities across eight countries, the WHO World Mental Health International College Student (WMH-ICS) project revealed that at least one-third of first-year university students reported one or more of the mental disorders examined in their survey, [16,17,23,69,70]. In Spain, the research by the Spanish Ministry of Universities [71] shows that between 49.5% and 52.8% of Spanish university students report anxiety and low levels of psychological well-being, and around 20% have insomnia and suicidal ideation. These data are corroborated by the research of Visier-Alfonso [72] and Ballester [44], who found high levels of stress in Spanish university students.

The present study considers mental health in line with characteristics outlined by WHO [73], i.e., more than the absence of mental disorders. In the present study, factors of perceived stress, suicidal behaviour, emotional intelligence, self-esteem and life satisfaction were considered within the evaluation of student mental health. Findings reveal that students with borderline or pathological symptomatology present greater perceived stress and suicidal behaviour, in addition to lower self-esteem and life satisfaction. 

In terms of emotional intelligence, it is important to differentiate between its three components. Firstly, attention to feelings pertains to the ability to feel and express feelings appropriately. Clarity consists of understanding one’s emotional state and, finally, repair describes the ability to appropriately regulate one’s emotional state [50,74]. Hence, emotional intelligence data collected in the present study reveal that students with better emotional intelligence in terms of emotional clarity and repair are more likely to exhibit normal emotional and behavioural symptomatology. In contrast, this group reports lower attention scores, probably because paying excessive attention to emotions can lead to psychological problems, emotional imbalances, physical symptoms, depression and anxiety. Furthermore, emotional intelligence is considered to be protective against stress, given that university students with better emotional intelligence present with lower perceived stress. In other words, improving emotional intelligence may prevent students from suffering perceived stress in higher education and, consequently, protect their personal well-being and self-esteem [75,76,77]. 

Further, students in the present study with better emotional outcomes exhibited healthier behaviours such as higher physical activity engagement. As has already been demonstrated in a number of previous studies, physical activity is a predictor of good mental and physical health [41]. In other words, higher levels of regular PA engagement exert a protective effect over mental and emotional health [40,78,79], although this effect reduces when individuals begin university because studying is the priority task for students, there is a progressive decrease in the weekly time devoted to physical activity and therefore a rise in sedentary time and stress levels [80,81]. 

Furthermore, present findings indicate that Mediterranean diet adherence is inversely associated with emotional and behavioural problems; that is, greater adherence is associated with fewer disorders. Corroborating research is found in studies conducted by Chen et al. [82] in the general population. Castro-Cuesta et al. [46] showed poor adherence to the MD in 54.9% of the Spanish university students. Vélez-Toral et al. [83] calculated that 29.9% of university students of Huelva had high adherence to the MD, and Ibáñez-del Valle et al.’s research [84] showed that poor adherence to the MD to be significantly and positively associated with depressive symptoms in nursing Spanish students. Other research in Spanish universities shows an average level of physical activity and adherence to the MD. However, these are small samples and from a single university degree, so it is not possible to generalise the results to Spanish students [81]. Nutritional interventions are one of the most widely applied, feasible and safe strategies used to promote health and functional capacity in individuals. Against this backdrop, encouraging a Mediterranean diet (MD) appears to be a sound scientific concept that is associated with favourable health outcomes throughout the life course [85,86,87]. Evidence indicates that the majority of university students follow an unhealthy diet. The present study shows that adhering to a Mediterranean diet produces similar outcomes to engaging in PA, with both being predictors of good health in university students. 

According to Pérez-Albéniz [21], students attending one university in northern Spain consume less tobacco, alcohol, and psychotropic drugs; however, they also demonstrated a significant increase in problematic internet use during the pandemic. In the present study, findings indicated that more compulsive internet use led to greater emotional and behavioural problems, which corroborates previous research [88]. Compulsive internet use (CIU) is not yet recognised by existing diagnostic classification systems, as there is still a lack of clarity regarding whether this behaviour should be considered a mental disorder or whether it is just a reflection of another underlying clinical condition [67]. Nonetheless, as is the case with alcohol consumption, it is clear that this behaviour is negatively associated with student well-being and a healthy lifestyle. This being said, the present study did not produce significant outcomes in the case of emotional and behavioural difficulties as a function of alcohol consumption in this sample of university students. Indeed, this finding is replicated in another research study [89].

Finally, the present study presents some limitations. Firstly, it reports an online survey conducted with a specifically selected sample and outcomes cannot be extrapolated to the general population, although they do paint a picture of the situation experienced by students attending a university in northern Spain. Secondly, the present study is based on self-report, for which usual limitations of self-report items, such as subjectivity-induced bias, must be taken into account. Despite this, employed instruments have previously been shown to be reliable and valid in previous studies with similar populations. Thirdly, the cross-sectional design of the present research impedes conclusions regarding causality from being reached. This should be addressed through future longitudinal studies. For future studies, the results suggest the need for longitudinal monitoring of common mental disorders among university students and of modifiable psychosocial and lifestyle factors that may contribute to ameliorating these mental health problems.

## 5. Conclusions

In summary, low perceived stress, suicidal behaviour and compulsive internet use, alongside high self-esteem, emotional intelligence and life satisfaction, physical activity engagement and Mediterranean diet adherence in university students are associated with fewer emotional and behavioural problems. In addition, the present study clearly underscores two issues. Firstly, emotional and behavioural problems are common amongst university students. Secondly, modifiable psychosocial and lifestyle factors are associated with concomitant outcomes for emotional and behavioural problems. 

The findings reported here can be used to inform the design of future actions targeted towards the interests and needs of university students informed by this preliminary contextualised evaluation. Universities have a key role in promoting and caring for the health of current and future society. For example, some Spanish universities have a healthy university service. Despite certain limitations in this study, including the reliance on self-reported health data, the use of a highly specific sample, and its cross-sectional design, future research should focus on developing personalized approaches to identify individual student risk profiles. Based on these profiles, appropriate intervention resources should be provided to mitigate the negative effects of mental disorders on this important segment of the population.

## Figures and Tables

**Table 1 healthcare-12-01482-t001:** Sociodemographic data.

		n	%
Age	Emerging adults (17–25 years)	1021	80.5
	Adults (>26)	247	19.5
Sex	Female	823	64.9
	Male	445	35.1
Employed	Yes	958	75.6
	No	310	24.4
Incomes	0–499	761	60
	500–999	203	16
	1000–1499	141	11.1
	1500–1999	71	5.6
	≥2000	92	7.3

**Table 2 healthcare-12-01482-t002:** Emotional intelligence and mental health-related behaviours according to the group of belonging obtained in the strengths and difficulties questionnaire.

	Strengths and Difficulties (SDQ)			*p*-Value
	Normal N = 918	Borderline N = 208	Abnormal N = 142	
Perceived stress	24.2 ± 7.76	32.84 ± 6.98	36.77 ± 6.98	<0.001
Suicidal behaviour	0.27 ± 0.78	0.75 ± 1.30	1.30 ± 1.59	<0.001
Self-esteem	33.01 ± 5.21	28.28 ± 5.24	24.75 ± 5.61	<0.001
Life satisfaction	18.12 ± 3.75	15.49 ± 3.36	13.63 ± 4.04	<0.001
Emotional intelligence: Attention to feelings	25.17 ± 6.64	27.06 ± 7.06	28.06 ± 7.61	<0.001
Emotional intelligence: Clarity of feelings	25.78 ± 6.34	22.10 ± 6.37	20.51 ± 7.09	<0.001
Emotional intelligence: Repair	26.44 ± 6.08	23.10 ± 6.31	20.53 ± 6.14	<0.001

**Table 3 healthcare-12-01482-t003:** PA engagement, toxics consumption and lifestyle habits as a function of strengths and difficulties symptomatology.

	Strengths and Difficulties (SDQ)			*p*-Value
	Normal N = 918	Borderline N = 208	Abnormal N = 142	
Physical activity (METs)	2880.91 ± 2389.28	2292.32 ± 2138.82	2229.23 ± 2243.51	<0.001
Sedentary time	389.19 ± 191.74	428.24 ± 211.06	437.48 ± 262.80	0.007
Mediterranean diet	6.32 ± 2.46	5.50 ± 2.52	4.80 ± 2.38	<0.001
Alcohol consumption	3.47 ± 3.44	3.66 ± 3.84	4.46 ± 4.78	0.587
Compulsive Internet Use	14.75 ± 9.98	20.27 ± 11.09	22.98 ± 11.71	<0.001

**Table 4 healthcare-12-01482-t004:** Correlation coefficients indicating relationships between strengths and difficulties questionnaire scores and all remaining examined variables.

	Perceived Stress	Suicidal Behaviour	Self-Esteem	Life Satisfaction	EI: Attention	EI: Clarity	EI: Repair	PA (METs)	Sedentary Time	MD ^1^	Alcohol Consumption	Compulsive Internet Use
SDQ	0.626 **	0.366 **	−0.578 **	−0.469 **	0.158 **	−0.353 **	−0.378 **	−0.133 **	0.085 **	−0.223 **	0.041	0.350 **

Note: *p* < 0.01 **. ^1^ MD: Mediterranean Diet.

**Table 5 healthcare-12-01482-t005:** Factors associated with emotional and behavioural difficulties.

	B	Std. Error	*p* Value	Adjusted R^2^
Perceived stress	0.221	0.017	<0.001	0.492
Suicidal behaviour	0.498	0.117	<0.001	
Self-esteem	−0.196	0.025	<0.001	
Life satisfaction	−0.090	0.034	0.009	
Emotional intelligence: Attention to feelings	0.044	0.017	0.010	
Emotional intelligence: Repair of feelings Emotional intelligence: Clarity	−0.062	0.020	0.002	
Mediterranean diet	−0.142	0.045	0.002	
Compulsive internet use	0.037	0.011	0.001	

Note: Only variables with statistically significant (*p* < 0.05) coefficients are presented. B: raw regression coefficient; Adjusted R^2^: coefficient of determination. Multiple linear regression model using the backward elimination method. Variables included in the model were: perceived stress, suicidal behaviour, self-esteem, life satisfaction, emotional intelligence (attention), emotional intelligence (clarity), emotional intelligence (repair), Mediterranean diet, physical activity, sedentary lifestyle, alcohol consumption and compulsive internet use (CIUS).

## Data Availability

Data is contained within the article.

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
