# Peer review of "Emotional and Behavioural Problems in Spanish University Students: Association with Lifestyle Habits and Mental Well-Being"

_healthcare, 2024, doi:10.3390/healthcare12151482_

Round 1

Reviewer 1 Report

Comments and Suggestions for Authors

The research work presented here responds to a concern for the scientific community and for this researcher. There is constant talk about the mental health of young people and we are grateful for the effort made in the research focusing on the mental health of young university students. With regard to the assessment of the same and, in accordance with the title, it is considered necessary to focus on the theoretical framework. It remains a justification without bringing the reader closer to key concepts in the work. In this way, it is surprising that the theoretical framework does not focus on welfare issues, but remains only a justification. The sample is very broad and an effort is made, but the type of sample or the profile of the participants is not explored in depth. It has been carried out in the north of Spain, but why is this not more a question of access to the sample? 

It is considered that the discussion of the results should be improved in depth and that, although the analyses are well founded, it is still a superficial analysis. Furthermore, the abstract states that the objective is: to analyse behavioural and emotional problems in university students. Abollas Duras and their relationship with psychosocial and lifestyle factors and, according to the results approach, it should be noted that they have focused on situating the presence of factors and competences and interrelating them, assuming that they directly or indirectly influence mental health.

We reiterate our thanks and support the authors in this scientific concern, but we consider that, in order to be published, the objective should be materialised and even focus the summary so that, when talking about mental health in students, we cannot understand the existence of emotional and behavioural problems as "normal". Hence, I consider that the abstract should be revised with a subject of such concern and interest.

Author Response

Comments 1: The research work presented here responds to a concern for the scientific community and for this researcher. There is constant talk about the mental health of young people and we are grateful for the effort made in the research focusing on the mental health of young university students. With regard to the assessment of the same and, in accordance with the title, it is considered necessary to focus on the theoretical framework. It remains a justification without bringing the reader closer to key concepts in the work. In this way, it is surprising that the theoretical framework does not focus on welfare issues, but remains only a justification.

Response 1: We appreciate the review comments. Following the received feedback, we have tried to align the introduction more directly with the title and objective of the manuscript, placing greater emphasis on well-being and defining emotional health.

Comments 2: The sample is very broad and an effort is made, but the type of sample or the profile of the participants is not explored in depth. It has been carried out in the north of Spain, but why is this not more a question of access to the sample? 

Response 2: Following the reviewer's suggestions, we have provided a more detailed description of the sample in Table 1. The term "northern Spain" has been used to help the reader locate the study population, and the specific name of the university is subsequently defined.

Comment 3: It is considered that the discussion of the results should be improved in depth and that, although the analyses are well founded, it is still a superficial analysis.

Response 3: Following the reviewer's suggestions, we have tried to provide a more in-depth discussion.

Comment 4: Furthermore, the abstract states that the objective is: to analyse behavioural and emotional problems in university students. Abollas Duras and their relationship with psychosocial and lifestyle factors and, according to the results approach, it should be noted that they have focused on situating the presence of factors and competences and interrelating them, assuming that they directly or indirectly influence mental health.

Response 4: We agree with the reviewer's comments. Accordingly, the abstract has been thoroughly revised.

Comment 5: We reiterate our thanks and support the authors in this scientific concern, but we consider that, in order to be published, the objective should be materialised and even focus the summary so that, when talking about mental health in students, we cannot understand the existence of emotional and behavioural problems as "normal". Hence, I consider that the abstract should be revised with a subject of such concern and interest.

Response 5: We appreciate the comments provided. We agree that the existence of these emotional and behavioral problems cannot be assumed to be "normal," and we have revised the abstract and the document to avoid this assumption.

Reviewer 2 Report

Comments and Suggestions for Authors

Dear Authors

It is an interesting work. However, the following suggestions should be taken into account to improve the quality of the manuscript:

- The type of methodology used is not mentioned in the abstract.

- Avoid acronyms in the keywords SQD (Strengths and Difficulties Questionnaire).

- Practitioner Points' is included under the heading which does not follow the structure required by the journal, review this section.

- Line 90, What the WHO acronyms mean

- In section 3.3 Tools, several questionnaires for collecting information are mentioned. However, none of them mention the degree of reliability of these questionnaires.

 - Check typographical characters

- The discussion is a dialogue between the findings and the work of other authors. Text citations are not recommended in this section, check this point.

- It is suggested that the references be revised according to the journal's standards, and also include the DOI in cases where it is possible, which is an interesting piece of information.

Best regards

Author Response

Comments 0: It is an interesting work. However, the following suggestions should be taken into account to improve the quality of the manuscript:

Response 0: We appreciate the review of the manuscript and the comments provided. We hope that the changes made meet the requirements.

Comments 1: The type of methodology used is not mentioned in the abstract.

Response 1: Following the guidelines outlined in the review, efforts have been made to clarify the methodology in the abstract.

Comments 2:  Avoid acronyms in the keywords SQD (Strengths and Difficulties Questionnaire).

Response 2: Understood, it has been reviewed.

Comments 3: Practitioner Points' is included under the heading which does not follow the structure required by the journal, review this section.

Response 3: The 'Practitioner Points' have been removed from the manuscript as it is not a section included in the journal's guidelines. We appreciate the feedback.

Comments 4: Line 90, What the WHO acronyms mean

Response 4: It has been reviewed.

Comments 5:  In section 3.3 Tools, several questionnaires for collecting information are mentioned. However, none of them mention the degree of reliability of these questionnaires.

Response 5: Thank you for the comment. We have incorporated data on their reliability.

Comments 6: Check typographical characters

Response 6: The manuscript has been thoroughly reviewed again to avoid these errors.

Comments 7: The discussion is a dialogue between the findings and the work of other authors. Text citations are not recommended in this section, check this point.

Response 7: Understood, it has been reviewed.

Comments 8: It is suggested that the references be revised according to the journal's standards, and also include the DOI in cases where it is possible, which is an interesting piece of information.

Response 8: Thank you for the comments. The references section has been thoroughly revised to comply with the standards required by the journal.

Reviewer 3 Report

Comments and Suggestions for Authors

Thank you for giving this opportunity to review the paper. It`s well-designed study with appropriate sample size. Some comments for improvement:

1. Name of the country should be added to the title.

2. Name of scales should be added to Abstract.

3. Results should be described more in Abstract with some statistics.

4. Key-words should be selected from MeSH.

5. Time and place of the study should be added to Abstract.

6. Status of Spanish students related to the topic should be reported briefly in Introduction and discussed in Discussion substantially.

7. Psychometric properties of used tools should be added. In addition, the tools should be ordered as subsectins.

8. Demographic variables should be presented in first paragraph of Results as well as a separate table.

9. Legends and caption of tables should be expanded for more clarity.

10. The manuscript needs exact proof-reading.

11. Suggestion for future studies should be added.

Comments on the Quality of English Language

Minor editing of English language required

Author Response

Comments 0: Thank you for giving this opportunity to review the paper. It`s well-designed study with appropriate sample size. Some comments for improvement:

Response 0: We appreciate the review of the manuscript and the comments provided. We hope that the changes made meet the requirements.

Comments 1: Name of the country should be added to the title.

Response 1: Following the reviewer's comments, the name of the country has been included in the title.

Comments 2:  Name of scales should be added to Abstract.

Response 2: The names of the scales have been added to the abstract.

Comments 3: Results should be described more in Abstract with some statistics.

Response 3: Efforts have been made to include more results in the abstract by adding brief statistical data.

Comments 4: Key-words should be selected from MeSH.

Response 4: It has been reviewed.

Comments 5:  Time and place of the study should be added to Abstract.

Response 5: Thank you for the comment. We have incorporated this into the abstract.

Comments 6: Status of Spanish students related to the topic should be reported briefly in Introduction and discussed in Discussion substantially.

Response 6: Following the reviewer's comments, efforts have been made to add studies involving Spanish university students.

Comments 7: Psychometric properties of used tools should be added. In addition, the tools should be ordered as subsectins.

Response 7: We agree, the requested data has been incorporated into the methodology.

Comments 8: Demographic variables should be presented in first paragraph of Results as well as a separate table.

Response 8: Thank you for the comments. A table with sociodemographic data of the sample has been included.

Comments 9: Legends and caption of tables should be expanded for more clarity.

Response 9: Improvements have been made in this regard.

Comments 10: The manuscript needs exact proof-reading.

Response 10: The manuscript has been fully reviewed again.

Comments 11: Suggestion for future studies should be added.

Response 11: A paragraph has been included to address this issue.

Round 2

Reviewer 1 Report

Comments and Suggestions for Authors

Dear Authors 

Thank you for the effort you have made to respond to the suggestions. Indeed, concern for the socio-emotional well-being of students is something that occupies me in my research career. I consider it to be improved for publication, but please review it before publication:

-The formatting of the table you have just incorporated. Table 1 Socio-demographic data (conform to the norms and aesthetics of the other tables).

- Check the bibliographic references as there are some typos.

-It would be necessary to state in the conclusions not only the prospective, but also the limitations of the study.

On this basis, I consider that the article should be published.

Best regards

Author Response

Comments1: The formatting of the table you have just incorporated. Table 1 Socio-demographic data (conform to the norms and aesthetics of the other tables).

Response1: Thank you for the comments. The table has been revised according to the format requested by the journal."

Comments2:  Check the bibliographic references as there are some typos.

Response2: The format of the bibliographic references has been thoroughly revised again. We hope that this time they meet the requirements

Comments3: It would be necessary to state in the conclusions not only the prospective, but also the limitations of the study.

Response3: Thank you for the comment. The main limitations have been added to the conclusions section (the limitations are described in more detail in the paragraph preceding the conclusions section)

Reviewer 2 Report

Comments and Suggestions for Authors

Dear Author/s

Please note the following comment after the appropriate review:

It is important to note that the suggestions from the previous peer review have been correctly incorporated into the manuscript, which significantly improves its quality. The abstract is clear in its description of the research.

The literature review has been extended with data aligned with the research, which provides a high degree of consistency to the research. The reliability indices of the questionnaires used to support this research have been included. The methodology is clear with the new information included. The discussion includes works by other authors that give support, clarity and good backing to the results obtained.

Reviewing a redundant sentence in lines 126 and 127 (the final sample...) is suggested. On the other hand, the format of the tables should be the same for all, therefore the format of Table 1 should be reviewed.

In conclusion, this article is interesting, provides important information and should be considered for publication in this journal.

Yours faithfully.

Author Response

Comments1: It is important to note that the suggestions from the previous peer review have been correctly incorporated into the manuscript, which significantly improves its quality. The abstract is clear in its description of the research.

Response1: We greatly appreciate the reviewer's confirmation.

Comments2: The literature review has been extended with data aligned with the research, which provides a high degree of consistency to the research. The reliability indices of the questionnaires used to support this research have been included. The methodology is clear with the new information included. The discussion includes works by other authors that give support, clarity and good backing to the results obtained.

Response2: We appreciate the confirmation of the improvements made in the various sections indicated by the reviewer on the previous occasion. We believe the article has improved based on the suggestions provided earlier.

Comments3:  Reviewing a redundant sentence in lines 126 and 127 (the final sample...) is suggested. On the other hand, the format of the tables should be the same for all, therefore the format of Table 1 should be reviewed.

Response3: The indicated sentence has been reviewed, and the format of Table 1 has been changed accordingly